

# Temporal and spectral cloud screening of polar-winter aerosol optical depth (AOD): impact of homogeneous and inhomogeneous clouds and crystal layers on climatological-scale AODs

Norman T. O'Neill[1], Konstantin Baibakov[1], Sareh Hesaraki[1], Liviu Ivanescu[1], Randall V. Martin[2], Chris Perro[2], Jai P Chaubey[1], Andreas Herber[3], Thomas J. Duck[2]

[1] Centre d'Applications et de Recherches en Télédétection, Université de Sherbrooke, Sherbrooke, Canada
[2] Dept. of Physics and Atmospheric Science, Dalhousie University, Halifax, Canada
[3] Alfred Wegener Institute for Polar and Marine Research, Bremerhaven, Germany

*Correspondence to*: Jai Prakash Chaubey (jaiprakash.spl@gmail.com)

**Abstract**

We compared star photometry-derived, polar winter aerosol optical depths (AODs), acquired at Eureka, Nunavut, Canada and Ny-Ålesund, Svalbard with GEOS-Chem (GC) simulations as well as ground-based lidar and CALIOP retrievals. The results indicate significant cloud and/or low-altitude ice crystal (LIC) contamination which is only partially corrected using temporal cloud screening. Spatially homogeneous clouds and LICs that remain after temporal cloud screening represent an inevitable systematic error in the estimation of AOD: this error was estimated to vary from 78% to 210% at Eureka and from 2% to 157% at Ny-Ålesund. Lidar analysis indicated that LICs appeared to have a disproportionately large influence on the homogenous coarse mode optical depths that escape temporal cloud screening. In principle, spectral cloud screening (to yield fine mode or sub-micron AODs) reduces pre-cloud-screened AODs to the aerosol contribution if one assumes that coarse mode (super-micron) aerosols are a minor part of the AOD. Large, low frequency, differences between these retrieved values and their GC analogue appeared to be often linked to strong, spatially extensive planetary boundary layer events whose presence at either site was inferred from CALIOP profiles. These events were either not captured or significantly underestimated, by the GC simulations. High frequency AOD variations of GC fine mode aerosols at Ny-Ålesund were attributed to sea-salt (SS) while low frequency GC variations at Eureka and Ny-Ålesund were attributable to sulfates. CALIOP profiles and AODs were invaluable as spatial and temporal redundancy support (or, alternatively, as insightful points of contention) for star photometry retrievals and GC estimates of AOD.

## 1 Introduction

The importance of understanding aerosol mechanisms driving the direct and indirect effects is of particular significance over the Arctic where climate change impacts are known to be amplified (IPCC, 2013). This is very important during the polar winter when aerosol variability, generally associated with the Arctic haze phenomenon, is typically stronger than during the polar summer (see Di Pierro et al., 2013 for example) and when the number and nature of Arctic haze aerosols can have significant (indirect) effects on thin ice cloud properties and their radiative forcing budget (c.f. Garrett & Zhao, 2006 and Blanchard et al., 1994 respectively).

In order to properly evaluate aerosol processes and emission representation in chemical transport models one needs to develop a reliable and varied measurement system to exercise as many of the aerosol functionalities as possible. Ground and satellite based remote sensing (RS) measurements are arguably the key components of such a measuring system since they provide the front-line, robust parameters that define the first order comparative constraints that models must necessarily satisfy. There are currently only a few instances of aerosol RS measurements during the polar winter: (a) satellite-based, polar orbit, lidar profiles and their derived aerosol optical depths (AODs) (b) ground-based lidar profiles and derived AODs as well as star photometer (and some moonphotometer) AOD measurements acquired at a few Arctic sites.





Star photometry is currently the defacto reference for all polar winter AOD measurements since it is a direct extinction measurement[1]. In the same way that RS parameters should be front line model comparison parameters, an AOD climatology should be a necessary basis of comparison in parallel to more spatially and temporally demanding (meteorological scale) evaluations. The AOD contamination impact of clouds and other sources of starphotometry error as well as the AOD computation impact of model limitations such as spatial resolution and time-step resolution are often dampened by carrying out comparisons at climatological scales.

In the Arctic, the process of cloud-screening raw star photometry AODs (of rejecting raw AODs, deemed to be cloud contaminated) is critical, given the relative weakness of AOD amplitudes as well as the occurrence of cloud and low-altitude ice crystal (LIC) events during the polar winter. Lesin's et al. (2009), studied LIC events at Eureka during 2006 and observed that 19.1% of lidar events were due to clear-night or cloudy-night LICs at an average altitude of $450 \pm 100$ m (average the Dec., Jan., Feb., March period of 2006). Cloud-screening may be temporal in nature (detected by rapid changes in optical depth where the assumption is that only clouds go through high frequency changes in optical depth) or of a spectral nature (ultimately based on the fact that cloud optical depths are spectrally neutral). The former approach suffers from errors of commission and omission (elimination of high frequency aerosol data and the inability to identify homogeneous cloud events respectively) while the latter approach may, for example, exclude super-micron aerosols (i.e. in addition to the cloud events which it is expected to exclude). If relevant comparisons are to be made with models then proper cloud screening is critical.

For our purposes, the current role of lidars in such climatologies is more of a supportive nature: ground-based lidars provide fundamental supporting data for AOD measurements in terms of the interpretation of the vertical contributions to the AOD (as well as the vertical contributions of cloud contamination) and the correlative coherence of their estimated AODs (Baibakov et al., 2015) while a satellite-based lidar provides critical interpretative information on the horizontal extent of these contributions and, their integrated AOD estimate.

High Arctic, near sea-level, star photometers at the AWI (Alfred Wegner Institute) base in Ny-Ålesund, Svalbard (79°N, 12°E) and the PEARL (Polar Environmental Atmospheric Research Laboratory) site at Eureka, Nunavut, Canada (80°N, 86°W) were employed to acquire a common, 2-year ensemble of polar winter AODs (Baibakov, 2014; Ivanescu et al., 2014). The simulated polar winter AODs of the GEOS-Chem (GC) model were compared with the star photometer AODs in order to quantitatively evaluate the relative temporal agreement of the star photometer and model over the 2-year reference period. Pan-Arctic AOD map products from the CALIOP lidar aboard the CALIPSO polar orbiting satellite (Winker et al., 2013). AOD animations for all daily orbit lines were compared with daily GC AOD maps to achieve a qualitative measure of the relative spatiotemporal agreement between the model and CALIOP animations and to better understand the extent of major AOD events during the polar winter.

## 2 Methodological considerations

In the text that follows we discuss specific issues related to the AODs derived from the measurements and model simulations. The symbol and acronym glossary allows for a centralized reference concerning the different types of AODs (whether measured or simulated) and other key parameters. As part of this study, we processed individual AODs and analyzed daily averaged and monthly averaged AODs.

### 2.1 Star photometer measurements

### 2.1.1 AODs generated by the star photometer

A brief description of the star photometer along with retrieval, calibration and logistical issues related to star photometer measurements is given in Baibakov et al. (2015). In that paper, we carried out an event level analysis of synchronized star photometer and Raman lidar measurements for a sampling of the data set employed in the present analysis. That communication was the first paper in which we reported on the optical coherency of passive / active, polar winter measurements subdivided into total, fine, coarse (optical) modes. It confirmed the relevance of extracting total, fine and coarse mode AODs ($\tau_a$, $\tau_f$ and $\tau_c$ at a reference wavelength of 500 nm) and motivated us to

---

[1] as opposed to the backscatter measurements provided by elastic and inelastic lidars which require, respectively, a knowledge of the transfer ratio from backscattering to extinction and an evaluation of the attenuation of the molecular signal.



create a preliminary AOD climatology which could be compared with AODs derived from GC simulations and CALIOP extinction profiles (see the symbol and acronym glossary for more details).

### 2.1.2 Spectral and temporal cloud-screening

As in Baibakov et al. (2015), raw AOD spectra were processed through the SDA (Spectral Deconvolution algorithm) to yield estimates of $\tau_a$, $\tau_f$ and $\tau_c$. The $\tau_f$ component is of particular relevance because it represents the contribution of aerosols that remain after the removal of the contribution of coarse mode clouds, coarse mode LICs
and coarse mode aerosols. This is what we call spectral cloud-screening: if the coarse mode aerosol contribution to $\tau_c$ is relatively small (and this is supported, for example by GC (aerosol) ratios of $\tau_{c,\,GC}$ / $\tau_{a,\,GC}$ being <~ 10% for the two stations across our climatological period) then one can argue that $\tau_f$ is representative of aerosols in the Arctic
and that $\tau_c$ is predominantly due to cloud or LIC contamination.
Baibakov et al. (2015) employed star photometry and lidar data to illustrate the utility of spectral cloud screening in the presence of temporally and spatially inhomogeneous clouds (their Fig. 8) as well as the effectiveness of both temporal and spectral cloud screening in the presence of inhomogeneous LICs embedded in what appeared to be a background environment of more homogeneously distributed LICs (their Figure 9). They noted that the two cloud-
screening approaches gave similar results in the presence of relatively inhomogeneous LICs while indicating that the remaining difference was arguably due to temporally (spatially) homogenous coarse mode particles (which, given the argument above, would be predominantly due to homogeneous LIC layers or homogeneous clouds).

If one divides temporally cloud-screened (accepted) and rejected raw AODs (and their derived SDA components) into two ("cs" and "rej") ensembles, then, for daily means (x = a, f, or c) the non cloud-screened AOD can be

divided into cloud-screened and non cloud-screened components

$$\tau_x \;=\; \gamma\,\tau_{x,\,cs} \;+\; (1-\gamma)\,\tau_{x,\,rej} \tag{1}$$


with $\gamma \;=\; N_{cs}/(N_{cs} + N_{rej})$ and where $N_{cs}$ and $N_{rej}$ are the number of AODs in each ensemble. Equation (1), one can be re-arranged to yield a sum of homogeneous and inhomogeneous components;


$$=\; \tau_{x,\,hom} \;+\; \tau_{x,\,inh} \tag{2}$$

where $\tau_{x,\,cs}$ has been renamed $\tau_{x,\,hom}$ in order to achieve a more intuitive vocabulary and where the inhomogeneous component (the perturbation above the low frequency, cloud-screened, homogeneous component) is,
$$\tau_{x,\,inh} \;=\; (1-\gamma)\left[\tau_{x,\,rej} - \tau_{x,\,hom}\right] \tag{3}$$

The fine mode AOD can be considered approximately homogeneous ($\tau_f \cong \tau_{f,\,hom}$; this is largely the basis of

temporal cloud-screening). Appealing to equation (2) and the propagation of $\tau_a = \tau_f + \tau_c$ across averages applied to any of the data ensembles (see the acronym and symbol glossary), $\tau_{f,\,inh} \cong 0$ and thus $\tau_{a,\,inh} \cong \tau_{c,\,inh}$. Equation (2) can then be expanded, for x = a;

$$\begin{aligned}\tau_a \;=&\; \tau_{a,\,hom} \;+\; \tau_{a,\,inh} \tag{4}\\ =&\; \tau_{f,\,hom} + \tau_{c,\,hom} + \tau_{f,\,inh} + \tau_{c,\,inh}\\ \cong&\; \tau_{f,\,hom} + \tau_{c,\,hom} + \tau_{c,\,inh} \tag{5}\end{aligned}$$

Equation (5) approximately represents the components of spectral cloud screening while, in comparison with

equation (4), reminds us that the cloud-screened AOD ($\tau_{a,\,hom}$) is divided into homogeneous components ($\tau_{f,\,hom}$ and $\tau_{c,\,hom}$) and that $\tau_{a,\,inh} \cong \tau_{c,\,inh}$. Equations (2), (4) and (5) propagate into monthly averages (maintain the same form).



## 2.2 GEOS-Chem simulations


The model that we employed for our comparisons was the GEOS-Chem global chemical transport model (GC) version 9-# (http://acmg.seas.harvard.edu/geos/). It is driven by GEOS-5 assimilated meteorological fields from the NASA Goddard Modeling and Assimilation Office (GMAO). The GC simulation has a 15 minute time step for transport and a 60 minute time step for chemistry and emissions. The lat / log grid size over the Arctic was 2° by
2.5° (approximately 220 km x 50 km respectively at the high Arctic latitudes of Eureka and Ny-Ålesund) with 47 vertical levels up to 0.01 hPa.
An overview of the aerosol physics and chemistry in GC is given in Park et al., (2004). We divided GC AODs into their fine and coarse mode components ($\tau_{f, GC}$ and $\tau_{c, GC}$) using the species by species segregation provided by GC (fine mode organic carbon, sulfate and black carbon along with fine and coarse mode sea-salt (SS) and mineral
dust). The GC aerosol simulation includes the sulfate-nitrate-ammonium system (Park et al., 2004; Pye et al., 2009), primary (Park et al., 2003) and secondary (Henze et al., 2006; Henze et al., 2008; Liao et al., 2007; Fu et al., 2008) organics, mineral dust (Fairlie et al., 2007), and sea salt (Jaegle et al., 2011). AOD is calculated at 550 nm using
RH-dependent aerosol optical properties (see Martin et al., 2003 for an overview of the optical processing employed for GC aerosols).

## 2.3 AODs generated from CALIOP profiles


The CALIOP processing algorithm generates attenuated backscatter coefficient profiles and, after the application of an aerosol classification algorithm, estimates of tropospheric AOD along a given CALIPSO orbit line. A discussion

of CALIOP extinction coefficient and AOD retrievals and their sources of variability within an Arctic night context can be found in Di Pierro et al. (2013). The AODs are, even in the significantly more optimal environment of nighttime conditions, very sensitive to the vagaries of aerosol vs cloud classification in conditions of weak
backscatter return typical of the relatively low concentrations of Arctic aerosols under or mixed with thin clouds or LICs, etc.. Di Pierro et al. (2013) suggest, for example, that sub 2-km "diamond dust" may have been misclassified as aerosols and thus may have been responsible for very high values of aerosol extinction coefficient (and thus of
AOD) from CALIOP retrievals (in 5% of the multi-year, December to February, Arctic-scale cases that they sampled).
With these considerations in mind, we employed CALIOP profiles and CALIOP AOD animations to gain insights into the spatio-temporal dynamics of aerosol events which might have influenced measurements at Eureka and Ny-Ålesund. We also employed averages of near-Eureka and near-Ny-Ålesund CALIOP AODs (i.e. spatial averages of
all CALIOP AODs within a specified radial distance from Eureka and Ny Alesund) as an auxiliary AOD context in our temporal comparisons of GC AODs with star photometer AODs at Eureka and Ny-Ålesund. We chose 500 km as the radius of the near-site CALIOP averages since this case generally displayed the least amount of day to day
variance in comparison with smaller radii choices (reduction in standard deviation of about a factor of 3 when increasing the radius from 100 to 500 km). The AODs were retrieved from the CALIOP "Column_Optical_Depth_Aerosols_532" product associated with the "5km Aerosol Profile".

### 2.3.1 Impact of differences in wavelength


For reasons of historical consistency we chose to retain the standard output wavelength that we employ for

starhotometry retrievals (500 nm), the 532 nm lidar wavelength of CALIOP and the 550 nm GC standard. As an indicator of the impact of these wavelength differences (for the case of the fine mode where the decrease from 500 to 532 to 550 nm would be at its largest), we performed a 2009 to 2011 survey of $\tau_f$ values for 5 Arctic AERONET
stations. The results indicated that the global 550 nm average was less than 0.01 below the global 500 nm average.

## 3 Results

### 3.1 GC and CALIOP spatial comparisons





Spatial comparisons between CALIOP and GC AODs were spotty at best. CALIOP sampling represents a rather extreme statistical challenge with generally modest signal to noise for the weak aerosol optical properties typical of the Arctic and strong cloud / LIC layer interference coupled with a highly irregular, spatial sampling grid. In spite of these limitations we frequently observed strong, spatially expansive, PBL[2] backscatter structures of low DR[3] that
were characterized as aerosol layers by the CALIOP processing algorithm. These structures were often not captured by GC in the sense that the simulated AOD amplitude was typically much smaller than the computed CALIOP AODs. Strong GC AOD events, on the other hand, are often unsupported by any CALIOP evidence simply because
the atmosphere in the region of interest is cloud dominated (although there can be relatively small, tantalizing windows of cloudless sky which suggest a, difficult to substantiate, spatial correlation between the model and the measurements).
**3.2 Climatological-scale analysis of star photometer AODs**
Figures 1 and 2 show star photometer and GC AOD comparisons for, respectively, daily averages at Eureka and Ny-Ålesund during the polar winters of 2010/2011 and 2011/2012. Each graph includes estimates of non cloud-screened AODs ($\tau_a$ in grey), cloud-screened AODs ($\tau_{a,\ hom}$ in black), fine mode AOD ($\tau_f$ in light red), filtered fine mode AOD ($\tau_f *$ in dark red) and GC-estimated fine mode AOD ($\tau_{f,\ GC}$ dark red dashes). $\tau_f *$ represents our best attempt
at producing climatological-scale AODs: to ensure the survival of only the most robust estimates of $\tau_f$, we allow ourselves the luxury of eliminating $\tau_f$ values for which $\tau_f / \tau_a < 0.3$ (for which the risk of errors due to residual cloud contamination is greatest).

        The most striking feature of these curves, in particular for Eureka, is the notable variation in the AODs, before and after temporal or spectral cloud screening. The cloud screening (in particular the $\tau_f *$ spectral cloud-screening) tends

to reduce magnitudes towards the $\tau_{f,\ GC}$ values. We have confidence in the $\tau_f *$ estimations based on our lidar / star photometer event level comparisons of Baibakov et al. (2015) and based on our detailed analysis of the diurnal variation of individual $\tau_f$ retrievals: in general the $\tau_f *$ values in Figures 1 and 2 that were significantly higher than
the $\tau_{f,GC}$ values were associated with robust and diurnally smooth variations of individual retrievals (see Fig. S1 of the supporting information for starphotometer illustrations of robust and moderately robust fine mode event).
High frequency variations of $\tau_{f,\ GC}$ for Ny-Ålesund (in particular the late winter variations of 2012 seen in Figure 2d) are predominantly due to fine mode SS aerosols associated with the yearly winter depression and strong winds southeast of Greenland (see, for example. Ma et al., 2008). It is noteworthy that virtually all large-amplitude, high
frequency variation of $\tau_{f,\ GC}$ at Ny-Ålesund is due to SS: outside of these peaks the dominant species is generally sulfate (an affirmation based on a component by component analysis of $\tau_{f,\ GC}$ values) . It is difficult if not impossible to demonstrate any degree of correlative agreement between the sparse $\tau_f *$ points and the high
frequency $\tau_{f,\ GC}$ spikes. Fig. S2 of the supporting information shows an example of apparent coherence between GC AODs (dominated by fine-mode, SS aerosols) and CALIOP AODs (the largest $\tau_{f,\ GC}$ peak of Fig 2d corresponds to the same day as this illustration). However such examples were frustratingly rare given the frequent appearance of
strong SS plumes in GC imagery (of which Fig. S2 is one of many examples): this is no doubt partly due to cloud-contamination of CALIOP profiles but it conceivably might also be GC overestimates of SS AODs. Attempts to relate $\tau_{f,\ GC-SS}$ to NaCl mass concentration measurements[4] acquired at the Ny-Ålesund, Zeppelin observatory (475
m.a.s.l.) were inconclusive in the sense of achieving any kind of significant correlation (and we note that no better correlation was achieved if GC mass concentrations at the Zeppelin elevation were employed instead of $\tau_{f,\ GC-SS}$).
A notable Ny-Ålesund star photometry feature was what appeared to be a continuity of strong $\tau_f *$ values from the last week in November, 2011 to the first week in January, 2012  where $\tau_f *$ was ~ 3 times the $\tau_{f,\ GC}$ values (Figure 2c). We believe that this difference is real because of the robustness of individual $\tau_f$ variations mentioned above and
because the CALIOP vertical profiles of this period were often dominated by strong PBL events of low DR. These vertical profiles were associated with spatially broad and robust $\tau_{a,\ CALIOP}$ features that were either not captured or

---

[2] Planetary boundary layer
[3] CALIOP depolarization ratio (see the symbol and acronym glossary for details)
[4] http://ebas.nilu.no/default.aspx, link provided by Ove Hermansen , 2015





significantly underestimated by the GC simulations (see two examples of these PBL events in Fig. S3 and Fig. S4 of the supporting information). The predominance of PBL aerosol events during the polar winter was, in particular, noted by Di Pierro et al. (2013) as part of their 6 year Arctic climatology using CALIOP profiles. A sampling of the CALIOP and GC vertical profiles for the event of Fig. S3 showed that GC appeared to capture the general vertical
form of the PBL feature but with $\tau_{a,\ GC}$ (largely sulphate dominated) values that were much weaker than the $\tau_{a,\ CALIOP}$ values. In this context of negatively biased $\tau_{a,\ GC}$ values, Di Pierro (2013) also found a negative, fine mode, polar winter GC bias and suggested that an important fine mode component during the polar winter (and
currently not included in GC) is dry SS particles that result from the sublimation of crystals from wind blown snow events.
A prominent Eureka event was the largest $\tau_f *$ value of Fig. 1a (Mar. 1, 2011). This corresponded to a strong value of $\tau_{a,\ CALIOP}$ and what appeared to be a spatially broad, PBL CALIOP event of low DR whose spatial continuity was inferred to be frequently hidden by higher altitude clouds. A second notable Eureka event was the largest $\tau_f$* value
of Fig. 1b (Mar. 29, 2012). This was a very stable fine mode event ($\tau_f \gg \tau_c$ with low frequency diurnal variation typical of aerosol events) which, however, only lasted for about 2½ hours (a duration which, at this late date of Mar. 29, is the result of the star photometer's inability to track stars in the presence of competitive or dominant, sunlight-
induced background radiance). CALIOP data did not support this strong value but the $\tau_{a,\ CALIOP}$ maps were very spotty with strong cloud contamination in the vertical profiles (and Eureka overpasses were all daylight overpasses so that the S/N advantages of the polar winter were largely lost at this late date).

      Fig. 3a shows monthly averaged starphotometer AODs ($< \tau_a >$) partitioned into grey and black $< \tau_{a,\ inh} >$ and $< \tau_{a,\ hom} >$ components (in support of equation (4)) as well as $< \tau_a >$ partitioned into $< \tau_{f,\ hom} >$, $<$

$\tau_{c,\ hom} >$, and $< \tau_{c,\ inh} >$ components (in support of equation (5)[5]. The need for temporal cloud screening (the significant amplitude of $< \tau_{a,\ inh} >$) relative to ($< \tau_{a,\ hom} >$ is evident (especially for Eureka). It is also evident that a significant fraction of homogeneous coarse mode values have circumvented the temporal cloud screening
process  dark blue ($< \tau_{c,\ hom} >$ values have been accepted as legitimate AODs). This (the unavoidable failure to reject raw AODs associated with homogeneous clouds or LICs) is a cloud / LIC detection error of the temporal cloud screening process (given, as indicated above,  the GC-driven assumption that coarse mode aerosols are a small
fraction of the AOD in the Arctic).  An estimate of the relative (%) error, due to this error of omission is $< \tau_{c,\ hom} > / < \tau_{f,\ hom} >$ : this yields values that range from 78% to 210% for Eureka and from 2% to 157% for Ny-Ålesund.

      In order to better understand the  large temporal cloud screening errors of the Eureka starphotometry data, we performed an analogous partitioning of lidar-derived coarse mode optical depths ($\tau_c'$) into inhomogeneous and

homogeneneous components above and below a nominal LIC upper limit ($h_{LIC}$ = 600 m using the statistical results of Lesins et al., 2013). The details of the partitioning process are given in Appendix B. The results, shown in Figure 3b, are colour coded to match the inhomogeneous / homogeneous colour coding of the Figure 3a starphotometry
results as well as being sub-divided into segments above and below $h_{LIC}$. The correspondence in terms of inhomogeneous and homogeneous partitioning is reasonable given the differences in sampling strategies of the two instruments as well as specific instrumental idiosyncrasies such as the overlap function associated with the lidar data
(see Appendix B for details). What is of particular interest is that the homogeneous contribution within the presumed LIC layer averages ~ 50% of the homogeneous total : a disproportionate amount in terms of vertical distance in the atmosphere (i.e. LICs appear to have an important influence on the homogenous coarse mode optical depths that
escape temporal cloud screening). At the same time we note the expected result that the inhomogeneous component is dominated by contributions above $h_{LIC}$.
Fig. 4a shows month to month variations, along with standard deviations of $<\tau_{a,\ CALIOP}>$,  $<\tau_f>$, $<\tau_f$*$>$ and, for the specific case of Ny-Ålesund, the monthly, 9-year star photometry climatology of Herber et al. 2002 ($<\tau_{a,\ Herber}>$). The variability (standard deviation) of $<\tau_{a,\ CALIOP}>$ is generally greater than the variability of spectrally cloud-
screened data ($<\tau_f>$ and $< \tau_f *>$). The differences in variability can be ascribed to differences due to orbit distance from our two sites, statistical anomalies due to the sparse and irregular nature of CALIOP AODs, and expected

---

[5] The fact that the two columns don't have the same height is a reflection of the approximate nature of equation (5) (that $< \tau_{f,\ inh} >$ is not negligible)





challenges in comparing two inevitably different methods of discriminating clouds and aerosols. The difference of $<\tau_f> - <\tau_f*>$ is generally small and positive with the biggest positive difference being ~ 0.03 for Eureka in March of 2011. $<\tau_f*>$ is ~ $<\tau_{a, Herber}>$ at Ny-Ålesund with certain months (Dec., 2011, Jan. 2012 and Mar., 2012) when it is significantly higher.

The standard deviations of Figure 4a aside, the estimates of $\tau_f$ and $\tau_f*$ are not all equal in terms of estimated SDA inversion errors. In Appendix B we show that the monthly averaged SDA retrieval errors ($< \Delta\tau_f >$) were inordinately large for the Ny-Ålesund data of 2011/2012 and that these large errors were associated with unphysically large spectral curvature values (large values of the monthly averaged 2$^{nd}$ derivative, $<\alpha'>$). While the retrieval errors were generally ~ the standard deviations for Eureka and the 2010/2011 season at Ny-Ålesund they were ~ 3 to 8 times the standard deviations of the 2011/2012 season.

Figure 4b shows a scale zoom (relative to Figure 4a) for the component selected for comparison with GC simulations ($< \tau_f*>$), alongside the $\langle \tau_{f, GC} \rangle$ predictions. The former is largely greater than the latter, in keeping with the results of Figure 1. The larger differences are frequently significant in terms of the standard deviations of the two data sets. These differences are most likely due to model underestimation, if only on the basis of the persistence of this apparent problem in the literature (Di Pierro, 2013; Breider et al., 2014). Potential sources of systematic bias in GC estimations could be ascribed to a missing fine mode component (such as Di Pierro's hypothesis concerning the lack of a modelled SS, fine mode aerosol ascribed to blown snow), emission underestimation, transport pathway errors, etc. Potential sources of systematic bias in the starphotometry estimates include the frequently sporadic temporal sampling of the star photometer as constrained by cloud and / or LIC conditions, acceptable levels of background sunlight in the late winter, star photometer calibration errors and errors in the SDA retrieval algorithm (there is also the wavelength difference bias, mentioned above, which would increase the $\langle \tau_{f, GC} \rangle$ values by $<\sim 0.01$ if those values had been computed at 500 nm). All measured and modelling cases in Fig. 4b, except for Eureka in 2011, show an increase from February to March. This increase is likely attributable to the late winter influence of Arctic haze (Herber et al., 2002) while the 2011 springtime increase in $\langle \tau_{f, GC} \rangle$ at Ny-Ålesund is primarily attributable to fine mode SS.

## 4 Conclusions

We performed a climatological-scale analysis of polar winter AODs measured at two high-Arctic sites in comparison with GC simulations and CALIOP retrievals. The results indicate significant cloud / LIC contamination which is only partially corrected with a temporal cloud screening algorithm. Temporal cloud screening eliminates raw AODs due to inhomogeneous (temporally and spatially variable) clouds and LICs. Homogeneous clouds and LICs that remain after temporal cloud screening represent an inevitable systematic error in the estimation of AOD which varies from 78% to 210% at Eureka and from 2% to 157% for Ny-Ålesund. In principle, spectral cloud screening (to obtain fine mode AODs) reduces raw AODs to the aerosol contribution if one assumes (supported by GC simulations) that coarse mode aerosols are a minor part of the total AOD. Lidar analysis indicated, for the case of Eureka, that LICs appeared to have a disproportionately large influence on the homogenous coarse mode optical depths that escape temporal cloud screening.

The SDA filtered parameter $< \tau_f*>$ was chosen as the most conservative approach for climatological-scale estimates of AOD. These values, typically larger than $\tau_{f, GC}$ estimates, are believed to be robust representations of $\tau_f$ variations: an important consideration in a context of weak amplitude and weakly varying signal embedded in an environment of large amplitude and strongly varying cloud and LIC signal. Large, low frequency, differences between $\tau_f*$ and $\tau_{f, GC}$ appeared to often coincide with strong PBL events whose presence at either site was inferred from spatially expansive, low-DR, PBL events in CALIOP profiles. These events were either not captured or, more likely, significantly underestimated, by the GC simulations. High frequency $\tau_{f, GC}$ variations at Ny-Ålesund were attributed to SS while low frequency variations at Eureka and Ny-Ålesund were attributable to sulfates. CALIOP profiles and AODs were invaluable as spatial and temporal redundancy support (or, alternatively, as insightful points of contention) for star photometry retrievals and GC estimates of AOD. Estimates of $< \tau_{a, CALIOP} >$ were found however to be significantly more variable than their fine mode counterparts from star photometry and GC simulations.


**Appendix A – lidar based partition into homogeneous and inhomogeneous coarse mode contributions**
Coarse mode optical depths ($\tau_c'$) derived from CANDAC Raman Lidar (CRL) profiles were computed as discussed in Baibakov (2015) for an ensemble of profiles characterized by a sampling interval of approximately 10 minutes. In a similar fashion to the homogeneous / inhomogeneous starphotometer AOD development above, the $\tau_c'$ values can be divided into homogeneous and inhomogeneous sub-ensembles. The process first involves, computing the
temporal derivative between pairs of $\tau_c'$ values and then discriminating homogeneous and inhomogeneous excursions by comparing the absolute value of each temporal derivative ($|d\tau_c'/dt|$) with a threshold value. This values was chosen to be 0.006 min$^{-1}$, the threshold discussed in Baibakov (2015) for star photometer sampling
intervals of approximately 5 minutes (although the actual threshold employed in that paper was strategically chosen to be roughly equivalent in performance to the 0.006 min$^{-1}$ threshold where the effective sampling interval was increased to an hour in order to better reject less inhomogeneous clouds). The lidar and star photometer were run
fairly independently during the 2010-2011 and 2011-2102 seasons and there was no strategic effort to have them collect synchronized data sets; the result was a certain amount of commonality in their acquisition periods but also periods when one or the other was making measurements alone. This yielded monthly average statistics for which $<$
$\tau_c >$ was significantly greater than the starphotometer average. We accordingly filtered the values with a maximum $\tau_c'$ cutoff filter so that their monthly average was equal to the starphotometer average ($<\tau_c>$) for each of the 4 months of Eureka data acquisition employed in our comparisons.

In a similar fashion to equation (1) above, monthly averages of $\tau_c'$ can be expressed as;

$$< \tau_c' >_- = \gamma_{hom-} < \tau_{c,\ hom}' >_- + (1 - \gamma_{hom-}) < \tau_{c,\ inh}' >_- \qquad (A1)$$

$$< \tau_c' >_+ = \gamma_{hom+} < \tau_{c,\ hom}' >_+ + (1 - \gamma_{hom+}) < \tau_{c,\ inh}' >_+ \qquad (A2)$$


for integrations below and above $h_{LIC}$ (the assumed upper limit of LICs). We note that these averages are carried out over individual lidar profiles and thus that there is no daily averaging (i.e there is no use of a bold font as in

equation (1)). The parameter $\gamma_{hom-}$ is given by $\gamma_{hom-} = N_{hom-}/(N_{hom-} + N_{inh-})$ where $N_{hom-}$ and $N_{inh-}$ are the number of coarse mode optical depths in the homogeneous (accepted) and inhomogeneous (rejected) sub-ensembles for integrations below $h_{LIC}$ (analogous expressions exist for the "+" case above $h_{LIC}$). We note that
the $\gamma$ factors are conservative ($\gamma_{hom\pm} + \gamma_{inh\pm} = 1$) because the total number of lidar-derived optical depths over the averaging period of a month;
$$N = N_{hom-} + N_{inh-} = N_{hom+} + N_{inh+}$$

(a given lidar-derived optical depth must be in one of the two sub-ensembles). The lidar-derived

average for the total profile is given by;

$$< \tau_c' > = \frac{\sum_{i=1}^{N} \tau_c'}{N} = \frac{\sum_{i=1}^{N}(\tau_{c-}' + \tau_{c+}')}{N}$$


$$= < \tau_c' >_- + < \tau_c' >_+ \qquad (A3)$$

Substituting equations (A1) and (A2) into (A3) yields;

$$< \tau_c' > = \gamma_{hom-} < \tau_{c,\ hom}' >_- + (1 - \gamma_{hom-}) < \tau_{c,\ inh}' >_-$$


$$+ \gamma_{hom+} < \tau_{c,\ hom}' >_+ + (1 - \gamma_{hom+}) < \tau_{c,\ inh}' >_+ \qquad (A4)$$

thus partitioning $< \tau_c' >$ values into their homogeneous and inhomogeneous components, below and above $h_{LIC}$.



**Appendix B - SDA retrieval errors**

An error model for all retrieved parameters of the SDA (in particular $\Delta\tau_f$) is given in O'Neill et al. (2003). Two important influences on $\Delta\tau_f$, at least within the context of an empirical analysis of Eureka and Ny-Ålesund star photometry retrievals are the amplitude of $\tau_f$ and the second derivative of $\tau_a$ ($\alpha'$). Both influences can be
approximated by a simple expression. In the first instance one has the pure differential in terms of $\tau_a$ and the fine mode fraction ($\eta = \tau_f/\tau_a$);
$$d\tau_f = \eta\, d\tau_a + \tau_a\, d\eta \qquad\qquad (B1)$$

Empirically one finds that rms errors associated with rms errors in the input AOD spectra are approximated by;

$$\Delta\tau_f \cong \tau_a\, \Delta\eta \qquad\qquad (B2)$$

The uncertainty $\Delta\eta$ is a strong function of the curvature at least for positive $\alpha'$ (which is generally true for cases where $\eta$ is reasonably large). Thus;
$$\Delta\tau_f \propto \tau_a\, \alpha' \qquad\qquad (B3)$$

In the presence of comparatively strong variations in $\alpha'$, $\Delta\tau_f$ will be roughly proportional to $\alpha'$. For the 13 monthly

averages of Figure 3c we obtained the results shown in Figure B1. Curvature values were excessive in the Ny-Ålesund data of 2011-2012 and this produced the quite large values of $<\Delta\tau_f>$ seen in the figure. These excessive values correspond to unphysical spectral AOD variations, involving spectral changes (often non-physical valleys
and peaks) which cannot be described by Mie theory. The second order spectral polynomial that we fit to AOD spectra before the application of the SDA tends to smooth out these artifactual variations but there will nonetheless be a residual influence.
**5 Acknowledgements**

We would like to thank NSERC for CCAR funding via the PAHA and NETCARE projects, the NSERC training program in Arctic Atmosphere Science as well as NSERC DG funding, the CSA, and the CFI for their financial

support. The contributions of the PEARL, AWI and CALIOP operations and processing staff are gratefully acknowledged.

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



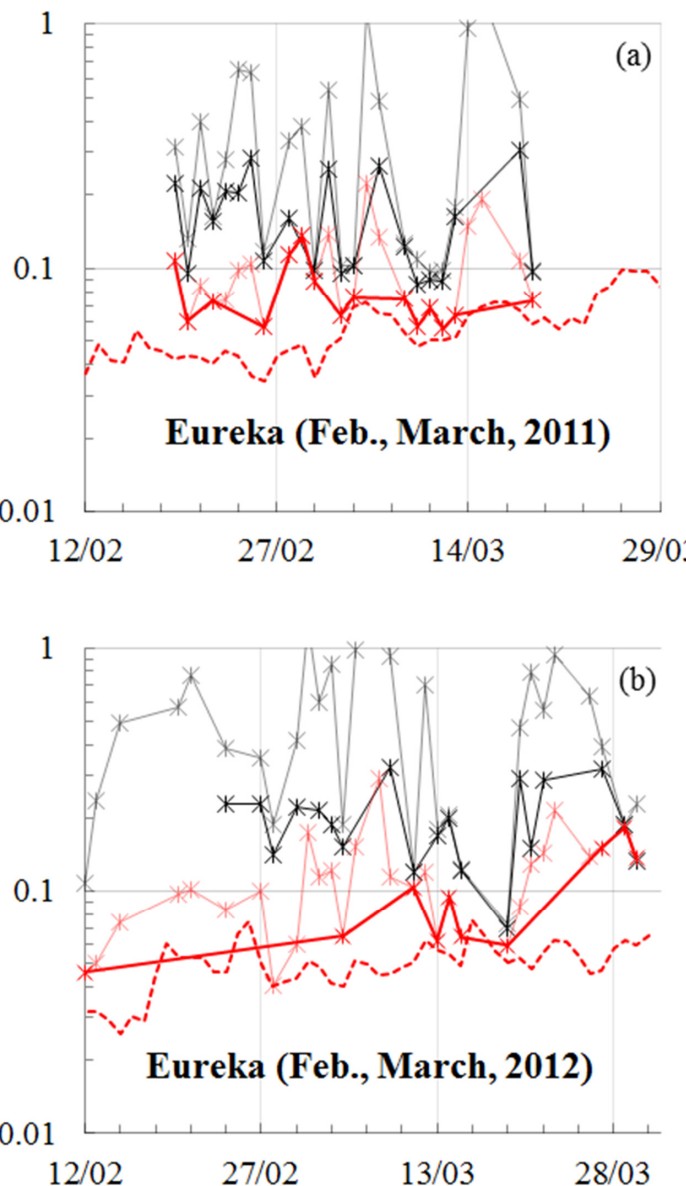


**Figure 1 -** Comparison of measured and cloud screened AODs (daily averages) derived from star photometry data with GEOS-Chem simulations over the polar winters of 2010/2011 and 2011/2012 at Eureka. The grey and black curves represent raw and cloud-screened AODs ($\tau_a$ and $\tau_{a,\,hom}$ respectively), while the light red and dark red curves represent the results of spectral cloud screening ($\tau_f$ and $\tau_f^*$ respectively). In order to be included in Fig. 1, all points required at least 10 raw AOD measurements per day. The simulated GC estimates of fine mode AOD ($\tau_{f,\,GC}$) are shown as dashed red curves (see nomenclature details in the symbol and acronym glossary).







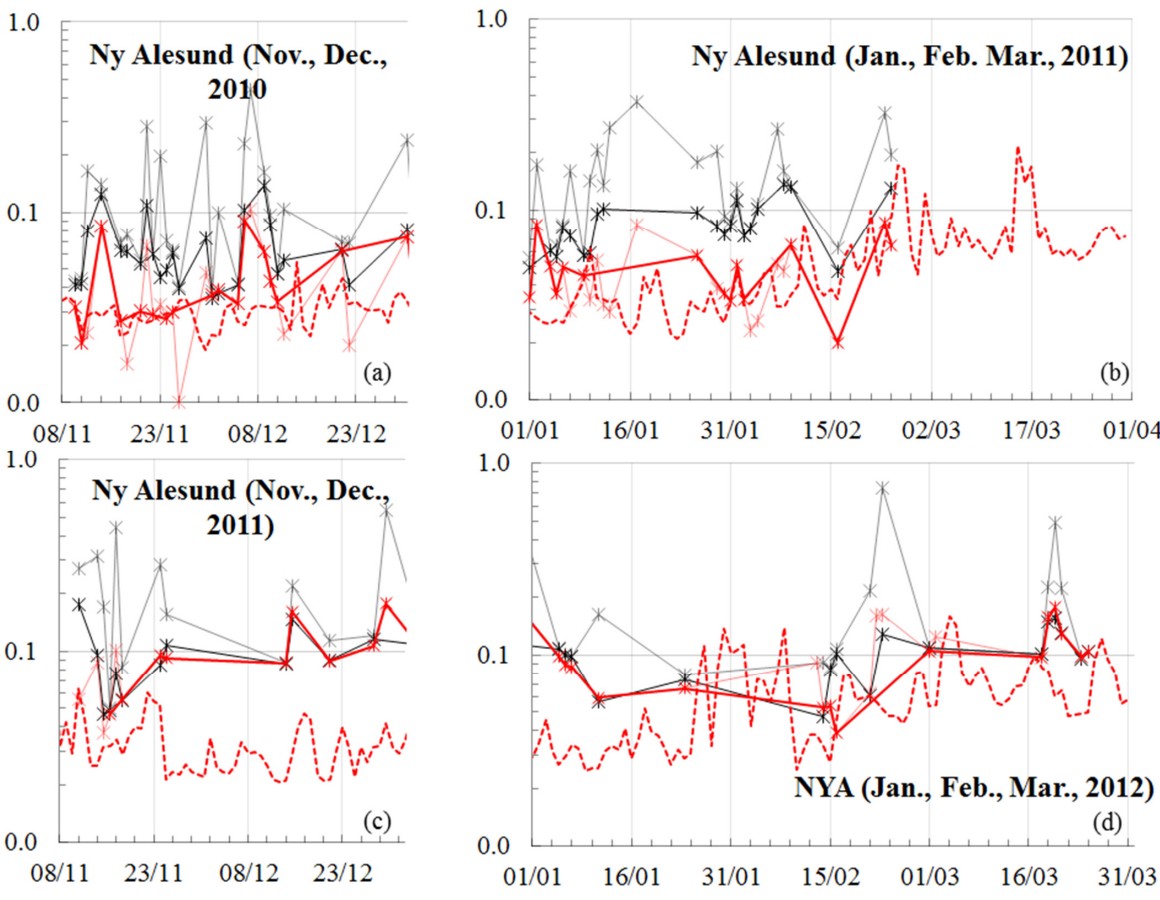


**Figure 2** As per the legend of Figure 1 but for Ny-Ålesund






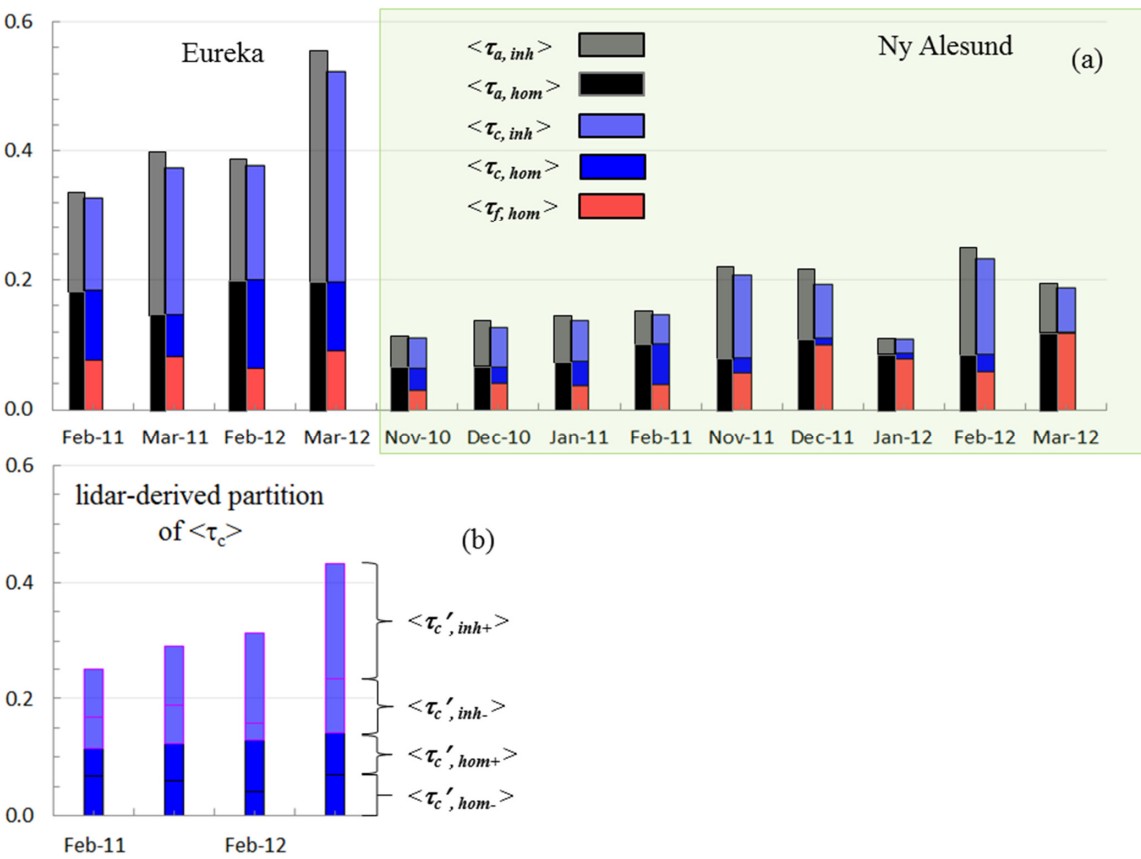


**Figure 3 - (a)** Comparison of temporal and spectral cloud screening (partitioned according to equations (4) and (5) respectively) for monthly AOD averages computed for Eureka and Ny-Ålesund during the polar winters of 2010/2011 and 2011/2012, **(b)** partitioning of lidar-derived coarse mode optical depths into homogeneous and inhomogeneous contributions above and below the nominal maximum altitude of low-altitude ice crystal layers ($h_{LIC}$) at Eureka.






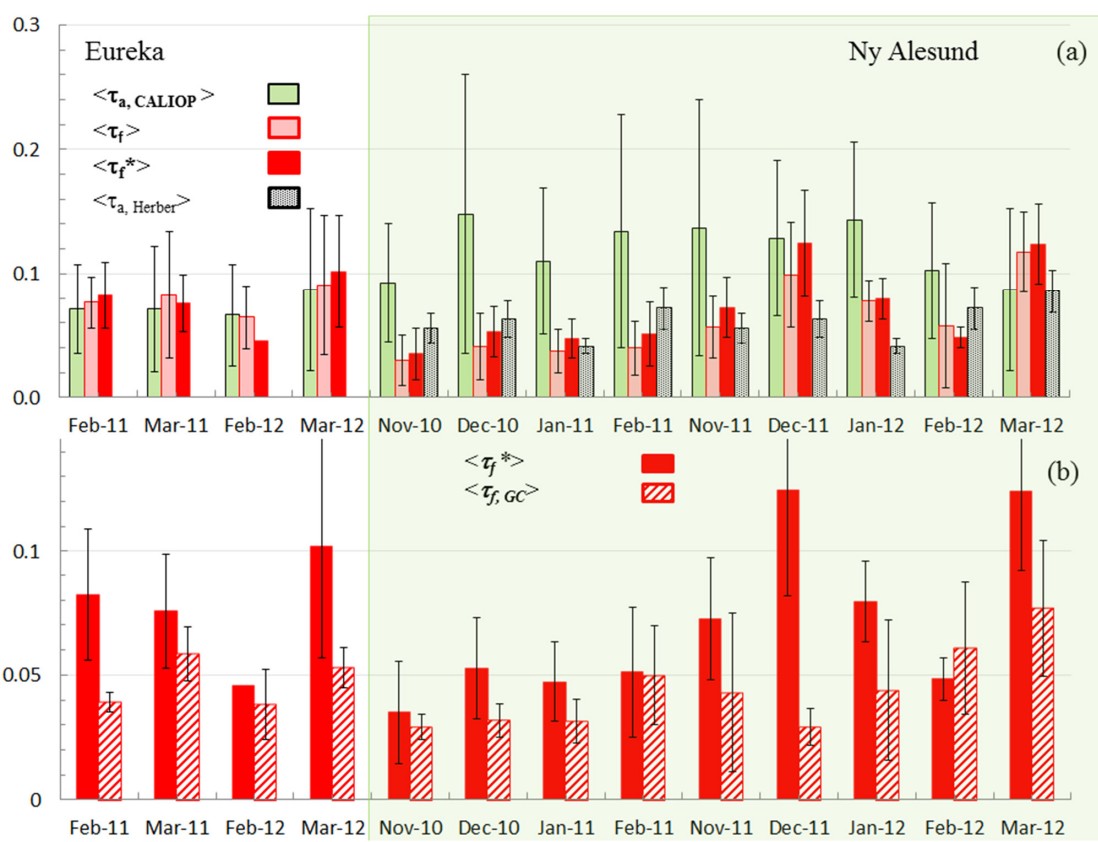



**Figure 4** - (a) $< \tau_{a,\,CALIOP} >$, $< \tau_f >$, $< \tau_f * >$, and the 9-year AOD climatology of Herber et al. (2002). For our purposes we simply repeated Herber's values that belonged to the same calendar month, **(b)** Zoom of the $< \tau_f * >$ values of (a) compared with

$< \tau_{f,\,GC} >$.





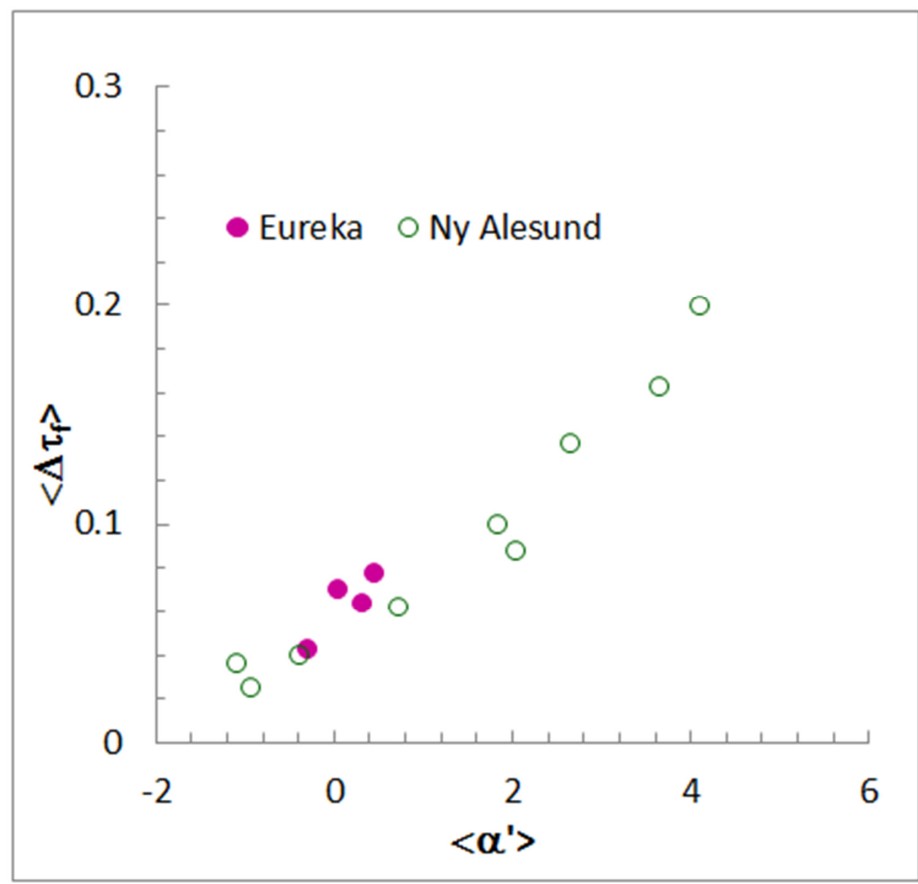


**Figure B1 :** Variation of the monthly averaged error in the SDA fine-mode AOD ($< \Delta \tau_f >$) as a function of the monthly averaged spectral curvature ($\langle \alpha' \rangle$, the second derivative of the spectral AOD; c.f. O'Neill et al., 2003 for details). These monthly averages were computed using individual measurements rather than daily averages (and thus $\Delta \tau_f$ and $\alpha'$ are not in bold)




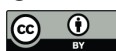

## Symbol and acronym glossary

### High-level definitions

| | |
|---|---|
| AOD | The community uses the acronym "AOD" to represent a variety of concepts. These range from nominal aerosol optical depth which hasn't been cloud-screened to the conceptual (theoretical) interpretation of aerosol optical depth. In this paper we use AOD in the latter sense and apply adjectives as required (see for example "raw AOD" below). |
| PBL | Planetary boundary layer. |
| raw AOD | Nominal AOD derived before temporal or spectral cloud screening. |
| SS | Sea-salt. |
| SDA | Spectral Deconvolution Algorithm : $\tau_x$ retrieval that employs AOD spectra as input (method described in O'Neill et al., 2003). |
| $\tau_x$ | $\tau_a$, $\tau_f$, or $\tau_c$ for total, fine and coarse mode AODs. Without explicit subscript qualification to another data source (CALIOP, GC, etc.) this nomenclature is reserved for outputs of the SDA (at a reference wavelength of 500 nm) applied to raw AOD spectra. $\tau_f$ or $\tau_c$ are conserved in the sense that $\tau_a = \tau_f + \tau_c$. This conservation expression, as well as its homogeneous, rejected and inhomogeneous components, propagates through daily and monthly averages. |
| $\overline{\tau_x}$ | Daily average of $\tau_x$ (in bold : this avoids an awkward nomenclature of $<<\tau_x>>$ for the monthly average of daily averages |
| $<\overline{\tau_x}>$ | Monthly average of $\overline{\tau_x}$. |
| UT | Universal Time : the time standard (with respect to 24 hour clock) employed throughout this study. |

### Lower level definitions

| | |
|---|---|
| DR | CALIOP Depolarization Ratio (see Winker et al., 2009 for a definition and a discussion on the particulate typing capabilities of the DR). |
| GC | GEOS-Chem, version 9.01.03. FlexAOD (Flexible AOD) is employed to perform offline calculations of AOD. |
| $h_{LIC}$ | assumed upper limit of LICs (assumed to be 600 m). |
| LIC | low altitude ice crystals. |
| $\tau_{x,\,cs}$ | $\tau_x$ values whose raw AOD inputs have been cloud-screened (have survived the cloud screening process). See $\tau_{x,\,hom}$ entry. |
| $\tau_{x,\,hom}$ | $\tau_x$ values associated with homogeneous conditions, defined as $\tau_{x,\,hom} = \tau_{x,\,cs}$ . |
| $\tau_{x,\,rej}$ | $\tau_x$ values whose raw AOD inputs were rejected by the temporal cloud-screening process (see Baibakov et al., 2015 for details). |
| $\tau_{x,\,inh}$ | $\tau_x - \tau_{x,\,hom}$ (only has meaning when it is computed from temporal averages of the cloud-screened and rejected points (from $\tau_{x,\,hom}$ and $\tau_{x,\,rej}$). |
| $\tau_{a,\,CALIOP}$ | Daily averaged (532 nm) values of the CALIOP AOD product within 500 km of Eureka / Ny-Ålesund. |
| $\tau_{f,\,GC}$ | Daily averaged GEOS-Chem, $\tau_f$ at 550 nm. In the supplementary information file, $\tau_{a,\,GC}$ values are used for comparisons with $\tau_{a,\,CALIOP}$ (i.e. since the CALIOP AOD product is not divided into fine and coarse mode contributions). |
| $\tau_{a,\,Herber}$ | AODs at 532 nm from the 10-year AOD climatology at Ny-Ålesund of Herber et al. (2002). Some simple interpolation was employed to estimate tropospheric AODs for months that were not given in their Table 3 (Oct., Nov., Jan. and Mar.). Total AOD values were computed by adding a stratospheric AOD of 0.01 (derived from the 525 nm case of their Figure 5). |
| $\tau_f^*$ | $\tau_f$ values on days for which $\tau_f / \tau_a$ values < 0.3 have been excluded. |
