# Peer review of "Temporal and spectral cloud screening of polar-winter aerosol optical depth (AOD): impact of homogeneous and inhomogeneous clouds and crystal layers on climatological-scale AODs"

_Atmospheric Chemistry and Physics, 2016_

## Referee Comment (RC1) · Anonymous Referee #2 · 7 Jun 2016

Final comments to the manuscript "Temporal and spectral cloud screening of polar-winter aerosol optical depth (AOD): impact of homogeneous and inhomogeneous clouds and crystal layers on climatological-scale AODs" submitted by O'Neil et al. to ACP.

The paper is dealing with very important issue related to the derivation of true AOD in wintertime using star photometers, and estimated to be worth while to be published in ACP.

[Figure]

However, there are several points to be modified before publication:

(General points) 1. There are too many acronyms, and some are not explained in the main text. Even you have a Table "Symbol and acronym glossary", you still need to explain in the text. You don't have explanation for "SDA", which is very important word in this paper, "DR" and "GEOS". What is GEOS? Also, you don't need to use some acronyms, such as SS or LIC.

2. "Spectral cloud screening" or SDA algorithm is not well explained, even might be described in some where else (in your PhD Thesis, Baibakov, 2014), it is still need to be shown in this paper.

3. Line 27-29 in abstract and line 245-252: Discussions of sea salt events might be compared with references not only of Ma et al., 2008, but also of many others.

(Specific points) 4. Line 137: "each ensemble" should be described as "cloud-screened" and "non cloud-screened".

5. Fig. 1: Why so many difference exists between the number of data points in cloud-screened AOD and spectral cloud screening results; grey, black, red and dark red?

6. Spectral cloud screening seems to be not well organized in case of Ny-Alesund because light red and dark red curves do not showing any substantial difference, especially in Fig. 2 (a), (c) and (d).

7. Generally, figures are not well referred in the main text.
* * *

---

## Referee Comment (RC2) · Anonymous Referee #3 · 10 Jun 2016

The paper present a study on the applicability of a cloud screening method to nocturnal star photometry AOD data. Even if of great importance in the analysis of these kind of data, it is not clear to me if this method can be easily applied to climatological time series. Another concern is that the language is too unformal at some points (e.g. at line 195 "Spatial comparisons between CALIOP and GC AODs were spotty at best" or at line 238 "A notable Ny-Ålesund star photometry feature was..").

Comments

Line 295: the difference you cite is not generally positive in my opinion (2 are positive and 2 negative for Eurika and generally negative for Ny-Alesund).

Appendix A: is not clear to me which is the need for this mathematical demonstration. Which is the physical sense?

Minor comments

Line 187: check "starhotometry"

Line 266: there is one "(" not necessary.

Line 278: should not be 'Appendix A'?

---

## Author Comment (AC1) · 26 Aug 2016

Reactions to the comments of referee 2 (referee comments italicized and bold, our reaction comments are neither italicized nor bold).

***Anonymous Referee #2***

***Final comments to the manuscript "Temporal and spectral cloud screening of polarwinter aerosol optical depth (AOD): impact of homogeneous and inhomogeneous clouds and crystal layers on climatological-scale AODs" submitted by O'Neil et al. to***

***The paper is dealing with very important issue related to the derivation of true AOD in wintertime using star photometers, and estimated to be worth while to be published in ACP.***

***However, there are several points to be modified before publication:***
***(General points)***

***1. There are too many acronyms, and some are not explained in the main text. Even you have a Table "Symbol and acronym glossary", you still need to explain in the text. You don't have explanation for "SDA", which is very important word in this paper, "DR" and "GEOS". What is GEOS? Also, you don't need to use some acronyms, such as SS or LIC.***
We already responded, in detail, and reacted with some changes in the text to this exact comment in our previous response to this reviewer (as part of the ACPD phase). In the absence of any kind of recognition of that response we can only presume that the reviewer didn't see our rebuttal : we accordingly didn't change anything in response to this comment since we believed that we had adequately responded in the previous revision phase.

***2. "Spectral cloud screening" or SDA algorithm is not well explained, even might be described in some where else (in your PhD Thesis, Baibakov, 2014), it is still need to be shown in this paper.***
As for point 1, we had already reacted to this exact comment (including text changes to the manuscript) in our previous response to this reviewer (as part of the ACPD phase). In this case the reviewer has inserted the text "(in your PhD Thesis, Baibakov, 2014)" into a copy of his previous comment. This insertion has no impact on the arguments we presented in our previous rebuttal (the PhD thesis gives a high level discussion of spectral cloud screening and the SDA while referring to (O'Neill et al., 2003) ; this is precisely the strategy that we pursued in our previous reaction to the reviewer).

***3. Line 27-29 in abstract and line 245-252: Discussions of sea salt events might be compared with references not only of Ma et al., 2008, but also of many others.***

***(Specific points)***

***4. Line 137: "each ensemble" should be described as "cloudscreened" and "non cloud-screened".***
Line 124 presumably. Made the replacement as suggested

We also attempted to clarify the whole narrative on the 3 data ensembles (raw or non cloud-screened, accepted or cloud-screened and rejected) in both the text surrounding equations (1) to (3) and in the Acronym and symbol glossary

**5. Fig. 1: Why so many difference exists between the number of data points in cloudscreened AOD and spectral cloud screening results; grey, black, red and dark red?**

Inserted the following parenthetical statement just after the first sentence of the second paragraph of Section 3.2: " (we leave the detailed discussion of these notable variations to the section below on temporal and spectral cloud-screening)". The "section below on temporal and spectral cloud-screening" refers to one of three new subsections in Section 3.2 whose delineation brings out a better focus on the key points that section (subsections called "Daily statistics", " Temporal and spectral cloud-screening", and "Monthly statistics"

**6. Spectral cloud screening seems to be not well organized in case of Ny-Alesund because light red and dark red curves do not showing any substantial difference, especially in Fig. 2 (a), (c) and (d).**
As for point 1, we already responded, in detail to this exact comment in our previous response to this reviewer (as part of the ACPD phase). In the absence of any kind of recognition of that response we don't see the point of reacting to this comment.

**7. Generally, figures are not well referred in the main text.**
We don't know what this means because we do refer clearly to all the figures in the text, in the Appendix and in the supplementary material. The reviewer would do well to illustrate such open ended comments with examples of where improvements could be affected

---

## Author Comment (AC2) · 26 Aug 2016

Reactions to the comments of referee 3 (referee comments italicized and bold, our reaction comments are neither italicized nor bold).

***The paper present a study on the applicability of a cloud screening method to nocturnal star photometry AOD data. Even if of great importance in the analysis of these kind of data, it is not clear to me if this method can be easily applied to climatological time series.***
Not sure what to do about this comment in terms of changes to the text. In all phases of the revisions of the manuscript we tried to play down the "climatological" aspect, recognizing that what we were providing was a "preliminary (testbed) AOD climatology" and to underscore that the main contribution to the paper is to point out the real problem of unfiltered homogeneous clouds.

***Another concern is that the language is too unformal at some points (e.g. at line 195 "Spatial comparisons between CALIOP and GC AODs were spotty at best" or at line 238 "A notable Ny-Ålesund star photometry feature was..").***
We'll live with the "spotty" (it is in Merriam-Webster) and we don't really see what is wrong with the line 238 (line 239).

***Comments***

***Line 295: the difference you cite is not generally positive in my opinion (2 are positive and 2 negative for Eurika and generally negative for Ny-Alesund).***
There was a mistake in the selection of the data for the graph ($< \tau_{a, \ hom} >$ was inadvertently plotted when it should have been $< \tau_a >$ that was plotted). The graph was corrected and now the statement we made is coherent with the graph.

***Appendix A: is not clear to me which is the need for this mathematical demonstration. Which is the physical sense?***
While equations (A1) and (A2) of Appendix A are analogous to equation (1), the rest of the derivation is necessary to formally show how the homogeneous and inhomogeneous lidar (coarse mode) optical depths are partitioned above and below $h_{LIC}$. The idea of partitioning above and below $h_{LIC}$ is critical to our argument about the importance of low altitude ice clouds and the derivation is needed to mathematically support that argument.

***Minor comments***
***Line 187: check "starhotometry"***
Fixed

***Line 266: there is one "(" not necessary.***
We could not find any unnecessary parenthesis

***Line 278: should not be 'Appendix A'?***
Yes : "Appendix B" replaced by  "Appendix A" (line 279)

---

## Author Comment (AC3) · 26 Aug 2016

Please find below the revised Figure 4.

[Figure]

---

## Author Response (AR2)

**Co-Editor Decision: Publish subject to technical corrections** (30 Aug 2016) by Dr. Matthias Tesche

*RS - omit the acronym and write out remote sensing on the very few occasions its used.*
OK, done

*GC - I recommend sticking to GEOS-Chem in the text, you can keep GC as index in the formulas.*
We looked at this in detail and, in our opinion, the employment of two acronyms for the model (one for the text and one for the variable subscripts) would confuse slightly more than it would help. We did however delay the definition of GC to the GEOS-Chem overview section (and replaced "GEOS-Chem global chemical transport model (GC)" by "GEOS-Chem global chemical transport model (hereafter referred to as GC)")

**SS - I think the readability of the text is better when you stick to sea salt.**
OK, done

*CALIOP and CALIPSO - acronyms are not introduced at all while others like AWI or PEARL are. Please be consistent.*
OK both acronyms are now defined where they appear in the abstract and the first time they appear in the text (but footnotes were used to do this; please see the comment immediately below)

*DR and PBL are introduced as footnotes which is rather unusual. Please put the full name in the text.*
Here is where we do not agree with commonly accepted editorial practice. The type of unwieldy, omnibus sentences that result from long-winded acronyms, detailed technical elaborations or a combination of the two should, in our opinion be minimized, whenever their inclusion excessively obscures the central message that one wants to get across. Expanding CALIOP and CALIPSO in one sentence is a very good example of an excessively unwieldy sentence[1]

*Also please make sure to point out if you are using the volume or particle depolarisation ratio.*
Done ("depolarization ratio" changed to "volume depolarization ratio" wherever it appeared)

*Finally, I think it should be mentioned what is meant with PBL event. Do you mean PBL development event?*
We added the following sentence to the PBL definition in the Symbol and acronym glossary; "We attribute no special meteorological significance to this term : rather we confine its usage to the empirical observation of low altitude, highly backscattering and horizontally extensive CALIOP profiles of low DR"
* * *
[1] Which would have been "Pan-Arctic AOD map products from the CALIOP (Cloud-Aerosol Lidar with Orthogonal Polarization) lidar aboard the CALIPSO (Cloud-Aerosol Lidar and Infrared Pathfinder Satellite Observations) polar orbiting satellite (Winker et al., 2013) were also used in this study."